# Laser-Based Selective Removal of EMI Shielding Layers in System-in-Package (SiP) Modules

**DOI:** 10.3390/mi16080925

**Published:** 2025-08-11

**Authors:** Xuan-Bach Le, Won Yong Choi, Keejun Han, Sung-Hoon Choa

**Affiliations:** 1Energy & Environment Research Institute, Seoul National University of Science and Technology, Seoul 01811, Republic of Korea; lexuanbach@seoultech.ac.kr; 2Genesem Inc., Incheon 21984, Republic of Korea; wychoi@genesem.com; 3School of Computer Engineering, Hansung University, Seoul 02876, Republic of Korea; 4Department of Semiconductor Engineering, Seoul National University of Science and Technology, Seoul 01811, Republic of Korea

**Keywords:** laser delamination, EMI shielding, system-in-package, selective removal, thermal stress analysis

## Abstract

With the increasing complexity and integration density of System-in-Package (SiP) technologies, the demand for selective electromagnetic interference (EMI) shielding is growing. Conventional sputtering processes, while effective for conformal EMI shielding, lack selectivity and often require additional masking or post-processing steps. In this study, we propose a novel, laser-based approach for the selective removal of EMI shielding layers without physical masking. Numerical simulations were conducted to investigate the thermal and mechanical behavior of multilayer EMI shielding structures under two irradiation modes: full-area and laser scanning. The results showed that the laser scanning method induced higher interfacial shear stress, reaching up to 38.6 MPa, compared to full-area irradiation (12.5 MPa), effectively promoting delamination while maintaining the integrity of the underlying epoxy mold compound (EMC). Experimental validation using a nanosecond pulsed fiber laser confirmed that complete removal of the EMI shielding layer could be achieved at optimized laser powers (~6 W) without damaging the EMC, whereas excessive power (8 W) caused material degradation. The laser scanning speed was 50 mm/s, and the total laser irradiation time of the package was 0.14 s, which was very fast. This study demonstrates the feasibility of a non-contact, damage-free, and selective EMI shielding removal technique, offering a promising solution for next-generation semiconductor packaging.

## 1. Introduction

Recently, advanced semiconductor packaging technology has emerged as a key approach to overcoming the limitations of Moore’s Law and enhancing the electrical performance of electronic systems. In particular, packaging technologies based on heterogeneous integration have gained significant attention [1,2]. Among these, system-in-package (SiP) technology has been widely adopted due to its advantages in achieving high integration density, superior electrical performance, compact form factor, and low cost [3,4]. In SiP technology, the semiconductor devices are integrated into a single package substrate and often integrated with various heterogeneous components such as antenna in package (AiP) [5,6]. With the integration of more chips into a single package, the chips are placed closer together, and electromagnetic interference (EMI) becomes one of the most critical concerns that need to be addressed [7]. The EMI-induced noise and crosstalk can degrade the electrical performance of the chips and damage electronic systems. Mitigating EMI is especially crucial in applications such as 5G RF modules, electric vehicles, and autonomous driving systems, where signal integrity and system reliability are important [8,9,10].

Traditionally, metal shield cans have been used to prevent electromagnetic interference (EMI) [11] and suppress noise emissions from electronic devices. However, their large and bulky form factors pose limitations when applied to advanced SiP modules that feature high chip density. To address these challenges, compartmental [12,13,14] and conformal shielding techniques [15,16,17] have been developed as package-level EMI mitigation solutions. Among these, conformal shielding has been widely adopted in SiP modules due to its compact size, low profile, simplified fabrication process, and excellent shielding performance. Various conformal shielding techniques have been developed, including spray coating [18,19,20], jet dispensing [21], electroplating [22,23], and sputtering methods [24,25]. Among these, sputtering has become the predominant technique for fabricating conformal EMI shielding layer. In this method, thin metal films—typically composed of copper and stainless steel—are deposited with micrometer-scale thicknesses onto the top and side walls of the epoxy mold compound (EMC) of the package to provide effective EMI shielding [26].

However, as semiconductor package structures become more complex and incorporate more electronic components, the number of areas where EMI shielding should not be applied is increasing. Certain regions within the package must remain uncoated or do not require the sputtered EMI shielding layer. Figure 1 shows an example of a SiP package that includes an AiP and electronic components that should not be coated. As shown in Figure 1, region A represents the antenna area (AiP), while region B corresponds to the electronic components or interconnection areas. In semiconductor packages for 5G technology, the number of antennas used in AiP or AoP (antenna on package) designs continues to increase. As a result, antennas originally placed at the edges of the package often need to be relocated toward the center of the package structure. To ensure proper signal performance, EMI shielding coating should not be applied to the top surface of the antenna area or electronic components.

Meanwhile, the sputtering process is a non-selective coating method, meaning the EMI shielding layer is uniformly deposited across the entire surface of the package. To overcome this limitation, a masking layer must be applied prior to sputtering to protect areas that should remain uncoated—such as antennas, electronic components, or interconnects—or alternatively, additional steps must be taken to remove the unwanted portions of the shielding layer after deposition. These extra steps increase process complexity, cost, and risk of damage to sensitive components. Even though the sputtering process using a mask is efficient for mass production, it is less suitable for complex package geometries and offers limited design flexibility. Recently, the selective patterning for EMI shielding layer was conducted using the polyimide masking tape and detaping process [27]. Although this technique is simple and low cost, it lacks precision and consistency, and suffers from the issues related to tape residue.

In this study, we propose a novel, laser-based technique for selectively removing the EMI shielding layer without the need for physical masking or post-processing. This approach is more flexible and particularly well-suited to complex SiP or AiP configurations. To assess the feasibility of this selective laser peeling method, numerical simulations were conducted to analyze thermal distribution and stress development during laser exposure—two key factors that influence delamination behavior and the risk of damage to underlying EMC materials. Based on the simulation results, a nanosecond pulsed fiber laser equipped with a line beam was employed in the experiments to evaluate the actual removal performance of the EMI shielding layer.

## 2. Test Vehicle and Numerical Modeling

Laser-based peeling of the EMI shielding layer is a relatively simple and intuitive method, well-suited for automation in semiconductor packaging processes. However, the laser may cause damage to the EMC mold material, and further optimization of the process is required. Therefore, prior to the experiment, a detailed numerical analysis was performed. This analysis aimed to assess the feasibility of selectively removing the EMI shielding layer using laser irradiation, determining optimal process parameters—such as laser power, scanning speed, and irradiation duration—and minimizing the risk of damage to the underlying EMC material. Figure 2 illustrates the test vehicle of the package structure, which is a simplified structure of the actual package, designed to evaluate the feasibility of the laser-based peeling process. The package was composed of a silicon chip, EMC material, and a PCB. It was assumed that the EMI shielding layer was deposited on the top surface of the EMC material. The EMI shielding layer had a three-layer structure composed of SUS (stainless steel)–copper–SUS, with the SUS layers each having a thickness of 1 μm and the copper layer 2 μm, resulting in a total thickness of 4 μm. The dimensions of the silicon chip, EMC, and PCB were 5 × 5 × 1 mm^3^, 7 × 7 × 2 mm^3^, and 16 × 16 × 1 mm^3^, respectively.

There are two possible laser irradiation methods for peeling the EMI shielding layer. The first method is full-area laser irradiation, as illustrated in Figure 2a. This approach utilizes a homogenized rectangular laser beam, which is commonly employed in laser bonding processes, also known as laser-assisted bonding techniques. The laser with the laser power of 20 W was directly irradiated onto the entire surface of the EMI shielding layer (7 × 7 mm^2^) for 1 s. The laser had a wavelength of 1064 nm, and the absorption coefficient of the SUS material at this wavelength was 34% [28]. The second method is the laser scanning method as shown in Figure 2b. In the laser scanning method, a line laser with the beam size of 1 × 7 mm^2^ was irradiated onto the EMI shielding layer and moved at a speed of 50 mm/s. The total irradiation time for the laser scanning method was 0.14 s, which is significantly shorter compared to the full-area laser irradiation.

A commercial ANSYS Workbench 2022 software was used to analyze the heat transfer and thermo-mechanical behavior of the EMI shielding layer and the package structure during laser irradiation. The absorbed laser energy causes a rapid temperature rise within the package, leading to the generation of thermo-mechanical stress due to the mismatch in the coefficients of thermal expansion (CTE) between the EMI shielding layer and the surrounding package materials. Uneven thermal expansion resulting from the CTE mismatch between the EMI shielding layer and the package materials, along with the thermo-mechanical stress at the interface, acts as the driving force for peeling off the EMI shielding layer from the EMC material.

Figure 3 illustrates the mesh structure of the finite element method (FEM) used in this study. To maximize accuracy, we employed Solid186 for the package structures, and SHELL181 element was used for the EMI shielding layer. The FEM model consists of a total of 28,365 elements and 126,268 nodes. We assumed that the package has no residual stress prior to laser irradiation, and all interfaces between the materials were considered to be perfect (with good contact) before laser irradiation. We did not account for the interface roughness or pre-existing micro-defects in the simulation modeling. Because the duration of laser irradiation is very short, the heat transfer process was analyzed by the transient thermal process A natural convection coefficient of 10 W/m^2^. K at an ambient temperature of 22 °C was applied. Following the laser irradiation, the results from the thermal analysis were used as input for the subsequent thermo-mechanical analysis of the entire structure.

For the thermo-mechanical simulation, the full package structure was modeled, with a fixed boundary condition applied at one corner of the bottom surface of the PCB. A stress-free temperature condition was set at room temperature (22 °C). The material properties used in the simulation are summarized in Table 1, and all materials were assumed to behave as linearly elastic.

## 3. Results and Discussion

### 3.1. Heat Transfer Analysis

During laser irradiation, the temperature changes in the EMI shielding layer and package materials are the primary factors inducing thermo-mechanical stress, leading to the delamination at the interface between the EMI shielding layer and the EMC material. Figure 4 presents the temperature distribution of the top surface of EMI shielding layer. As shown in Figure 4a, under full-area laser irradiation, the temperature rapidly rises to 238 °C within 1 s. In contrast, for the laser scanning method shown in Figure 4b, a significant temperature gradient is observed across the EMI shielding layer. At the beginning of the laser scanning process, the temperature of the EMI shielding layer is approximately 65 °C, while by the end of the irradiation (after 0.14 s), the temperature in the irradiated region reaches 221 °C.

Figure 5 illustrates the temperature distribution on the cross-section of package structure. In the case of full-area irradiation (Figure 5a), the maximum temperature of EMC and the silicon chip was 238 °C and 23 °C, respectively. Figure 5b shows the variation in the maximum temperature of the EMI shielding layer during laser irradiation. The temperature of the EMI shielding layer rapidly increased to 238 °C within 1 s and then quickly dropped to approximately 50 °C. Since the melting point of the EMC material is 270 °C, this temperature rise does not cause melting of the EMC material. Figure 6a,b show the temperature distribution of the package structure and the temperature profile of the EMI shielding layer under the laser scanning method. As shown in Figure 6b, the temperature of the EMI shielding layer sharply increased to 221 °C at the final stage of irradiation (after 0.14 s). The corresponding temperatures of the EMC material and the silicon chip were 221 °C and 22 °C, respectively. Therefore, neither of the laser irradiation methods caused melting of the EMC material or damage to the silicon chip. Notably, in the case of laser scanning irradiation, the high temperature was concentrated in a small, localized region, which can lead to local deformation. This localized deformation is expected to promote easier delamination of the EMI shielding layer compared to full-area laser irradiation.

### 3.2. Thermo-Mechanical Stress Analysis

It is expected that during laser irradiation, the highest thermo-mechanical stresses will develop at the interface between the EMI shielding layer and the EMC material, primarily due to the mismatch in the coefficients of thermal expansion (CTE) between the EMI shielding layer and the surrounding package. These thermo-mechanical stresses and resulting deformations are the main driving forces for delamination of the EMI shielding layer. Among these stresses, interfacial shear and tensile stresses play particularly important roles in initiating and propagating the peeling of the EMI shield layer.

Figure 7 illustrates the distribution of shear stress (τ_xy_) at the interface for two different laser irradiation techniques: full-area irradiation and laser scanning irradiation. The maximum shear stress was located at the corners of the EMI shielding layer, as indicated by the red and blue regions in Figure 7. Notably, the shear stress generated using the laser scanning method is significantly higher than that generated using the full-area irradiation method. In particular, the maximum shear stress in the case of laser scanning reaches 38.6 MPa, nearly three times higher than that observed in the case of full-area irradiation (12.5 MPa), despite the peak temperature during laser scanning being 17 °C lower. This is attributed to the localized heating effect during laser scanning, which induces concentrated thermal gradients and local deformation, resulting in significantly higher shear stress at the interface.

In this study, we qualitatively evaluated the adhesion strength between the EMI shielding layer and the EMC material before laser irradiation using the pencil tests and cross-cut tester. A total of five samples were tested. For the pencil tests, all samples showed 9H hardness. For the cross-cut tests, all samples achieved 5B level based on the ASTM D3359 standard [29] which showed an excellent adhesion strength of EMI shielding layer. There was no peeling-off and delamination of the layer after pencil and cross-cut tests. However, these are qualitative adhesion tests. In order to obtain the adhesion strength value of the EMI layer on EMC material, the shear tester should be used. However, the adhesion strength between the EMI shielding layer and the EMC material could not be measured experimentally, primarily due to the extremely low thickness of the EMI shielding layer, which makes the use of conventional shear testing methods unsuitable. Consequently, it was difficult to directly verify whether the stress generated during laser irradiation was sufficient to induce delamination. However, several previous studies have reported that the average adhesion strength between EMC materials and metal coatings such as nickel [30] and copper [31] typically ranges from 8 MPa to 10 MPa. Moreover, adhesion strength is known to decrease significantly with increasing temperature. Therefore, the calculated shear stress of 38.6 MPa observed during laser irradiation is considered sufficient to delaminate the EMI shielding layer from the EMC material.

In addition to shear stress, the tensile stress acting normal to the EMI shielding layer and EMC interface—commonly referred to as the peeling stress (σz)—is also a critical factor influencing interfacial delamination behavior. As shown in Figure 8, the maximum peeling stress was observed at the corners of the package. The calculated maximum peeling stresses were 9.6 MPa for full-area laser irradiation and 2.2 MPa for laser scanning irradiation, both significantly lower than the corresponding shear stresses. This is attributed to the fact that thermal expansion during laser irradiation occurred primarily in the horizontal (XY, in-plane) direction rather than in the vertical (Z, out-of-plane) direction, indicating that shear stress is the primary driving force for delamination. Based on the stress simulation results, delamination is expected to initiate at the package corners and propagate inward.

Figure 9 illustrates the distribution of the shear and peeling stresses at the onset of delamination at the package corners during laser scanning irradiation. A sharp increase in both stress components is observed in this region. Notably, in the transition zone between bonding and debonding areas, the maximum shear stress reaches 392 MPa, approximately 50 times higher than the typical adhesion strength between the EMI shielding layer and the EMC material. Additionally, the maximum peeling stress reaches 215 MPa. These results suggest that once delamination begins at the corners, it is likely to propagate rapidly inward due to the high interfacial stress concentrations.

### 3.3. Warpage Analysis

In this section, we investigated the deformation behavior and warpage of the top surface of the EMI shielding layer and the underlying EMC material. The simulation assumed that delamination of the EMI shielding layer initiates at the corners during laser irradiation, leading to partial debonding in those areas. As the temperature of the EMI shielding layer increases, the delaminated regions are expected to propagate inward. In the simulation model, regions without delamination were defined as having bonded contact with the EMC material, whereas the delaminated regions were modeled using a frictional contact condition with a friction coefficient of 0.2. Figure 10a shows a cross-sectional view of the deformation behavior of the EMI shielding layer and package structure during full-area laser irradiation. The EMI shielding layer and the package were deformed in a convex shape (︵). Figure 10b displays the warpage calculated on the diagonal direction of the EMI shielding layer and the top surface of the EMC material. The warpage of the EMC surface was 7.5 μm, while that of the EMI shielding layer was 5.6 μm. The resulting warpage difference between EMI shielding layer and EMC material was 1.9 μm, as indicated by the gap shown in Figure 10a.

Figure 11 also shows the deformation of the EMI shielding layer and package structure for the laser scanning irradiation. The warpage value of the top surface of EMC is 5.7 μm, and the warpage value of the EMI shielding layer is 2.3 μm. The difference in warpage value between the EMI shielding layer and EMC material was 3.4 μm, which is larger than 1.9 μm observed for full-area laser irradiation. This result suggests that the laser scanning method is more effective in promoting delamination of the EMI shielding layer compared to full-area irradiation.

### 3.4. Experimental Results

Following the numerical analysis, selective peeling tests for the EMI shielding layer using a laser were conducted. Figure 12 shows a photographic image of the semiconductor package used in the experiment. The package was coated with the EMI shielding layer on the EMC material. The size of the package was 4 × 20 mm^2^. The EMI shielding layer consisted of a multilayer structure of SUS–Cu–SUS, with SUS and Cu having a thickness of 1 μm and 2 μm, respectively—matching the dimensions used in the numerical analysis. Based on the numerical simulation results, the laser scanning irradiation method was found to be more effective for peeling off the EMI shielding layer compared to full-area laser irradiation. Therefore, only the laser scanning method was employed in the experimental tests. A nanosecond pulsed fiber laser (IPG Photonics, Oxford, MA, USA) with a line beam was used in the experiment. The width of the laser line beam was 25 μm, and the wavelength was 1064 nm. In the numerical simulation, a laser power of 20 W and a line beam width of 1 mm were used. However, the laser beam width was much narrower (25 μm) in the experiment. Therefore, to maintain the same heat flux under experimental conditions using a narrower 25 μm beam, the laser power was proportionally reduced to a range between 4 W and 8 W, as determined through heat flux calculations.

Figure 13 presents several optical images of the package after laser irradiation under a laser power of 4 W. When a low laser power of 4 W was applied, the EMI shielding layer either remained intact or peeled off only partially. At a laser power of approximately 6 W, the EMI layer was completely removed, as illustrated in Figure 14. Following the peeling-off test, the surface of the EMC was examined using an optical microscope, and no damage to the EMC material was observed. These results indicate that the laser-generated heat effectively removes the EMI shielding layer without causing damage to the underlying EMC material. To prevent contamination, the peeled-off EMI films and debris were continuously removed and collected using an air blower during the process. When the laser power was increased to 8 W, both the EMI shielding layer and the EMC material were damaged and degraded due to excessive heat, as shown in Figure 15. These results indicate that optimizing laser power and irradiation time is crucial for achieving effective and selective laser peeling of the EMI shielding layer. Although the present study focuses on the removal process and the short-term integrity of the exposed EMC surface, the long-term mechanical and chemical stability of the EMC after EMI shielding layer removal remains to be investigated. Future work will include accelerated aging tests and environmental exposure studies to evaluate the reliability of the exposed EMC under various conditions.

In conclusion, this study successfully demonstrated the feasibility and effectiveness of a laser-based selective peeling-off process for removing multilayer EMI shielding structures without damaging the underlying material.

## 4. Conclusions

In this study, we proposed a novel laser-based technique for the selective removal of EMI shielding layers in advanced semiconductor packages. Through detailed numerical simulations and experimental validation, the laser scanning method was shown to generate higher interfacial shear stresses and more localized thermal deformation compared to full-area laser irradiation, thereby promoting more effective delamination of the EMI shielding layer without damaging the underlying EMC material. The thermo-mechanical simulations revealed that the shear stress at the interface during laser scanning irradiation could exceed 38 MPa—well above the typical adhesion strength of metal coatings on EMC—while the peak temperatures remained below the melting point of EMC. Warpage analysis further supported the conclusion that localized heating induced by laser scanning results in greater delamination between layers. Experimental results using a nanosecond pulsed fiber laser confirmed the simulation results. Complete removal of the EMI shielding layer was achieved at optimized laser power levels (approximately 6 W), while higher power levels (8 W) led to EMC degradation, highlighting the importance of precise control over process parameters. The outcomes of this study are expected to contribute to the advancement of non-contact, selective EMI shielding layer removal methods which can significantly enhance advanced packaging processes in next-generation semiconductor manufacturing.

## Figures and Tables

**Figure 1 micromachines-16-00925-f001:**
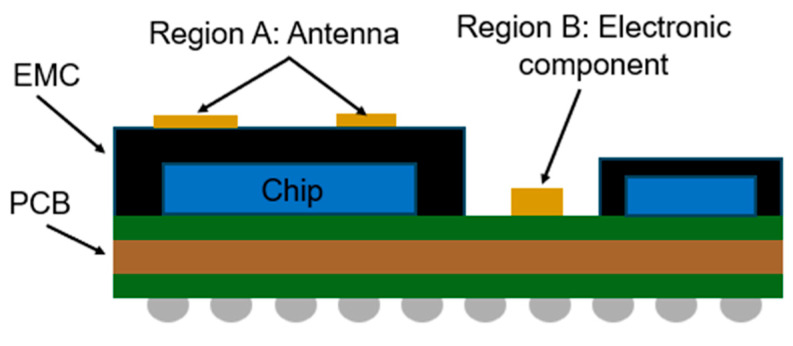
Schematic drawing of the SiP package including AiP and electronic components.

**Figure 2 micromachines-16-00925-f002:**
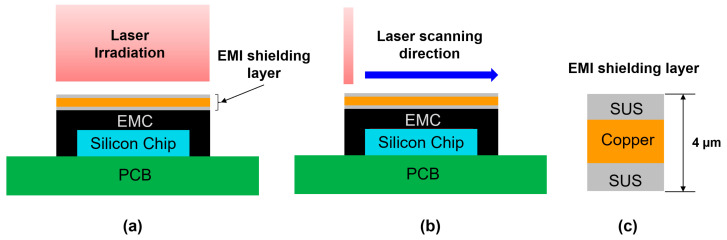
Schematic illustration of the laser irradiation methods: (**a**) full-area laser irradiation; (**b**) laser scanning irradiation; (**c**) structure of EMI shielding layer.

**Figure 3 micromachines-16-00925-f003:**
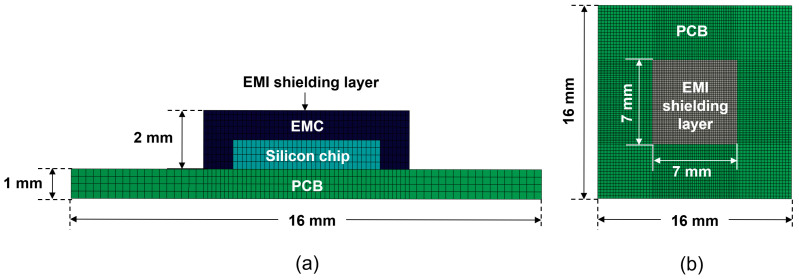
FEM mesh of the package structure: (**a**) cross-sectional view; (**b**) top view.

**Figure 4 micromachines-16-00925-f004:**
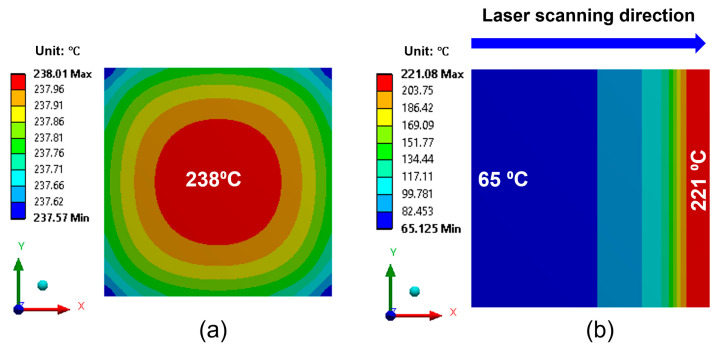
Temperature distribution of the EMI shielding layer after laser irradiation: (**a**) full-area laser irradiation; (**b**) laser scanning irradiation.

**Figure 5 micromachines-16-00925-f005:**
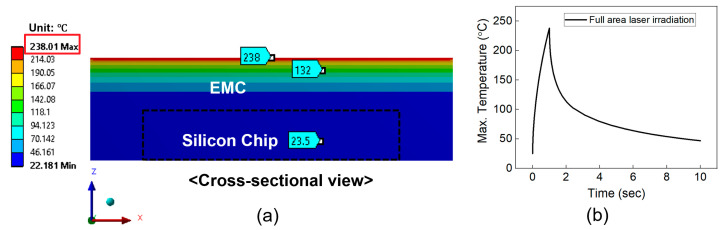
(**a**) Temperature distribution on the cross-section during full-area laser irradiation. (**b**) Maximum temperature profile of the package during full-area laser irradiation.

**Figure 6 micromachines-16-00925-f006:**
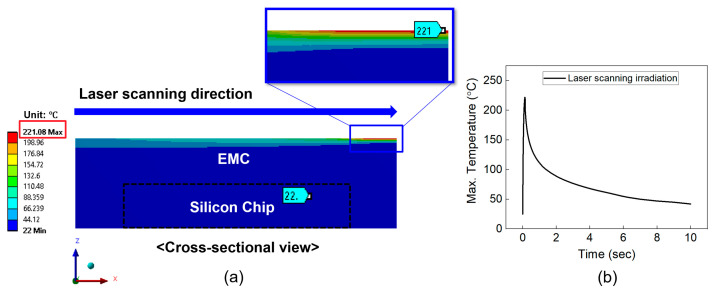
(**a**) Temperature distribution on the cross-section during laser scanning irradiation. (**b**) Maximum temperature profile of the package during laser scanning irradiation.

**Figure 7 micromachines-16-00925-f007:**
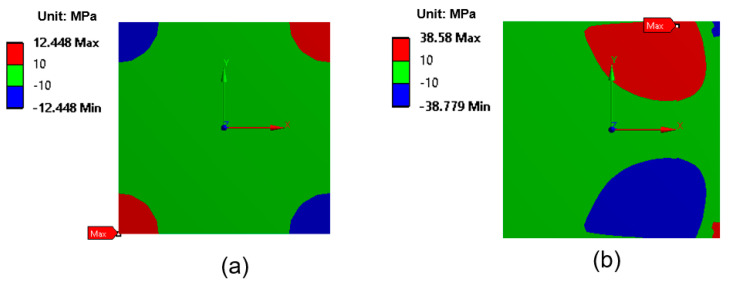
Shear stress (τ_xy_) distribution at the interface between the EMI shielding layer and the EMC: (**a**) full-area laser irradiation; (**b**) laser scanning irradiation.

**Figure 8 micromachines-16-00925-f008:**
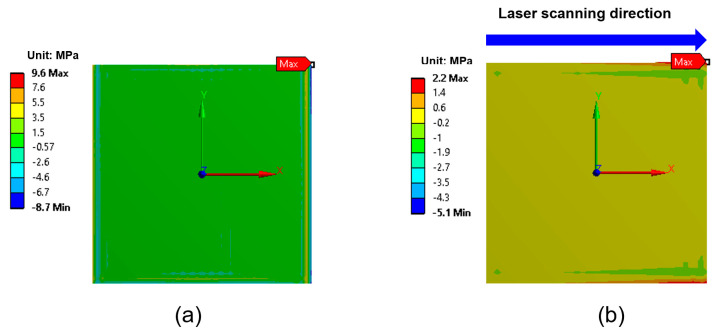
Peeling stress (σ_z_) distribution at the interface between the EMI shielding layer and the EMC: (**a**) full-area laser irradiation; (**b**) laser scanning irradiation.

**Figure 9 micromachines-16-00925-f009:**
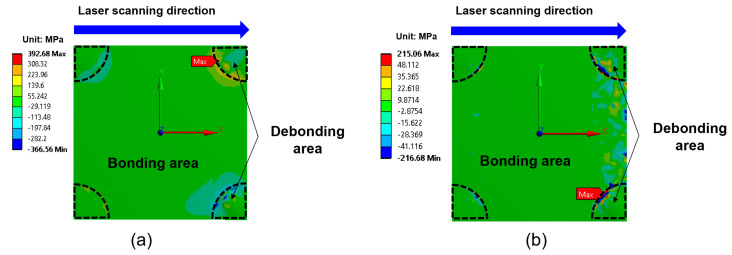
Stress distributions at the interface between the EMI shielding layer and the EMC during laser scanning irradiation when debonding occurs at the corners: (**a**) shear stress (τ_xy_); (**b**) peeling stress (σ_z_).

**Figure 10 micromachines-16-00925-f010:**
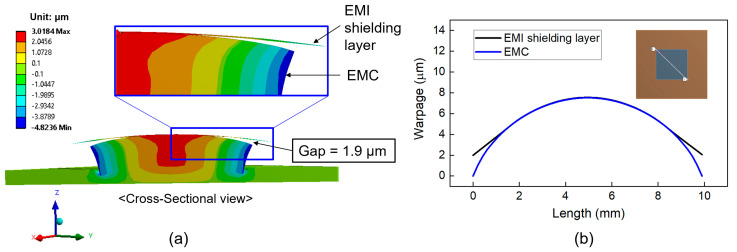
Warpage (displacement in the z-direction) of the package structure during full-area laser irradiation: (**a**) cross-sectional view showing warpage difference (magnified 100 times); (**b**) warpage profiles of the EMI shielding layer and EMC on the diagonal direction.

**Figure 11 micromachines-16-00925-f011:**
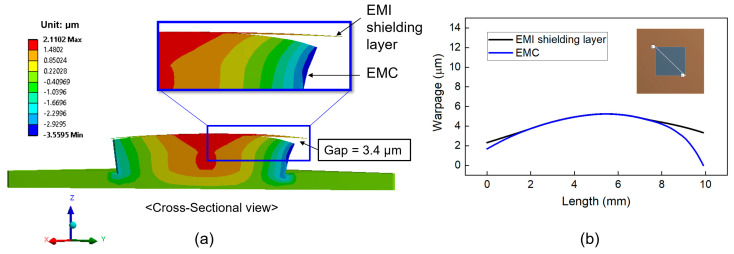
Warpage (displacement in the z-direction) of the package structure during laser scanning irradiation: (**a**) cross-sectional view showing warpage difference (magnified 100 times); (**b**) warpage profiles of the EMI shielding layer and EMC on the diagonal direction.

**Figure 12 micromachines-16-00925-f012:**
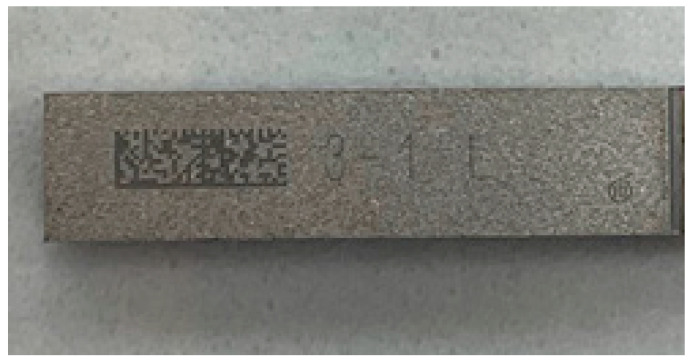
Picture image of the semiconductor package coated on EMI shielding layer.

**Figure 13 micromachines-16-00925-f013:**
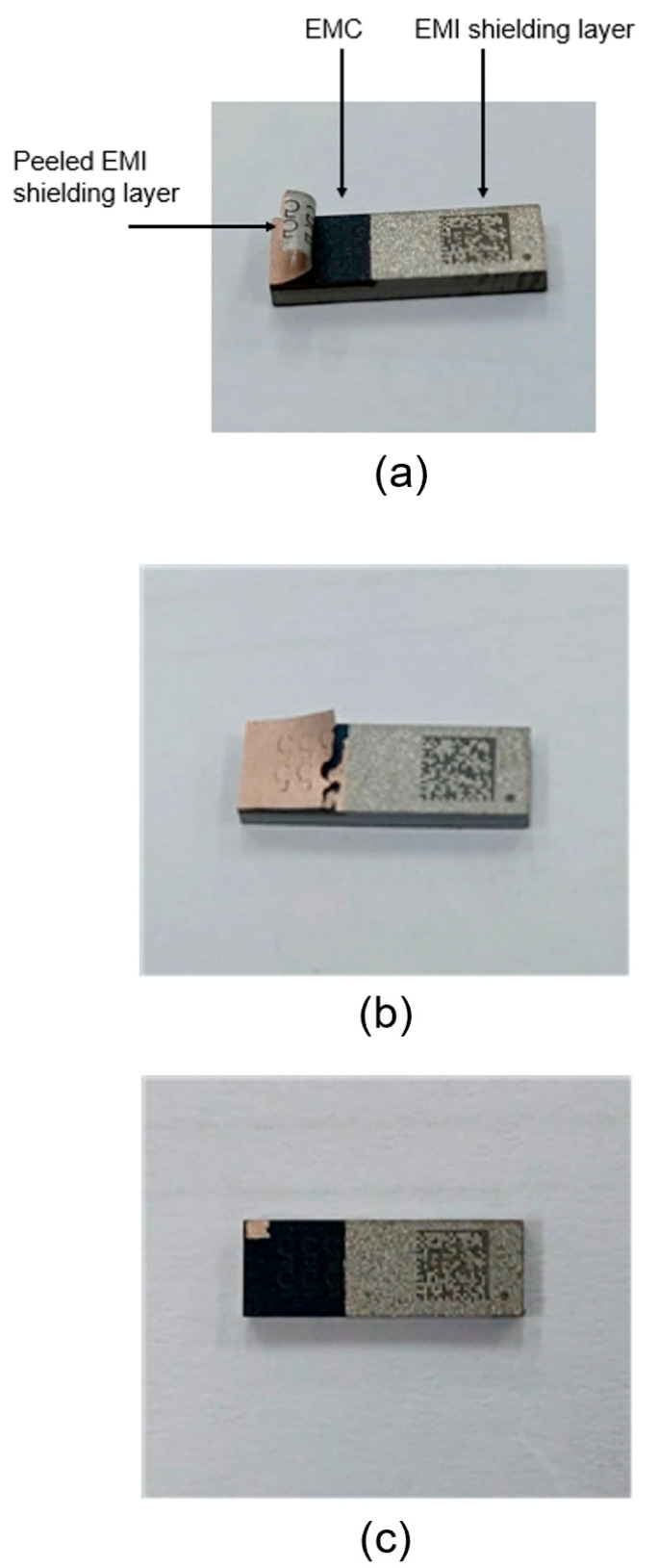
Optical images of the EMI shielding layer after laser irradiation at different power levels: (**a**) partial peeling-off at 4 W; (**b**) complete removal at 6 W; (**c**) damage to EMI and EMC layers at 8 W.

**Figure 14 micromachines-16-00925-f014:**
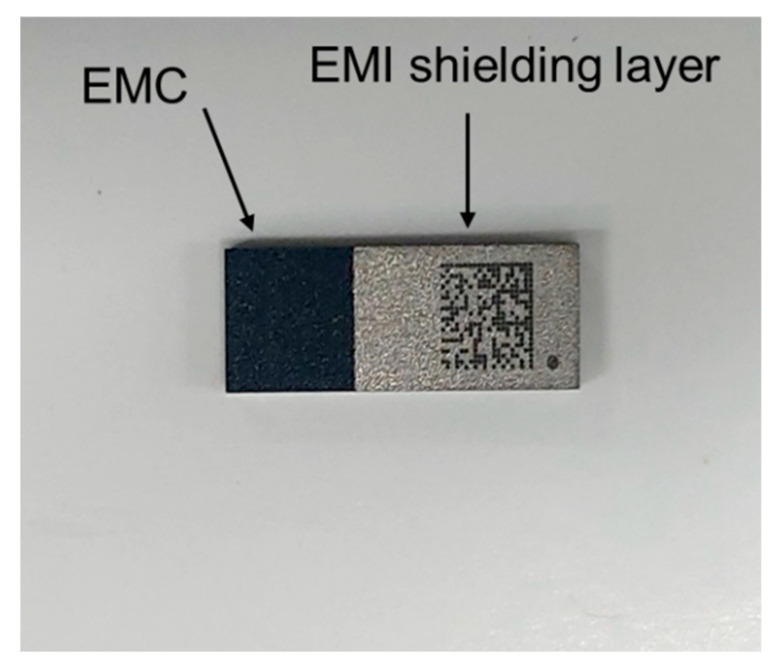
Optical image of the EMI shielding layer after laser scanning irradiation at 6 W, showing complete removal of the EMI shielding layer without damage to the underlying EMC.

**Figure 15 micromachines-16-00925-f015:**
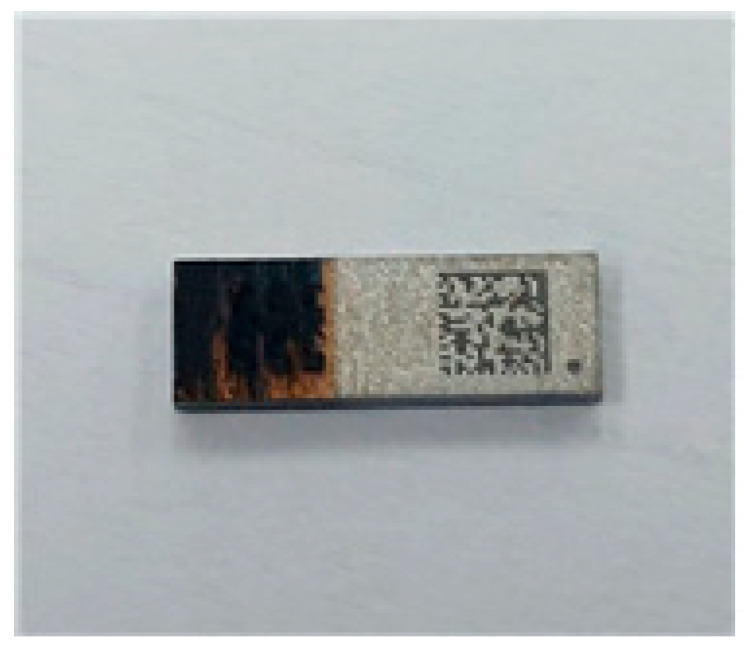
Optical image of the EMI shielding layer after laser irradiation at 8 W showing damage and degradation of both the EMI shielding layer and the underlying EMC material.

**Table 1 micromachines-16-00925-t001:** The properties of materials used in this paper.

Materials	Density (kg/m^3^)	*E* (GPa)	ν	α (ppm/°C)	Thermal Conductivity (W/m·K)	Specific Heat (J/kg·K)	Melting Point (°C)
Silicon	2300	131	0.28	2.8	148	794	1412
EMC	1800	19	0.23	10	0.25	1100	270
Copper	8960	128	0.34	16.5	400	390	350
SUS 304	8000	193	0.29	17.8	16.2	500	1400
PCB	1850	24	0.118	17	K_xy_ = 27.4	396	-
					K_z_ = 0.35		

*E* is Young’s modulus, *ʋ* is Poisson’s ratio, *α* is coefficient of thermal expansion.

## Data Availability

Data is contained within the article.

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
