# Peer review of "Laser-Based Selective Removal of EMI Shielding Layers in System-in-Package (SiP) Modules"

_micromachines, 2025, doi:10.3390/mi16080925_

Round 1
Reviewer 1 Report
Comments and Suggestions for Authors
The manuscript presents a promising laser-based method for selective EMI shielding layer removal. The simulations and experiments are well aligned, and the topic is relevant to advanced semiconductor packaging. However, the following key points should be addressed to strengthen the study. Therefore I recommend minor revision.
- The proposed method is compared qualitatively to traditional masking. However, a quantitative comparison of factors like cost, processing time, throughput, or yield would significantly strengthen the claim that this approach is advantageous. Consider including a comparison table or brief analysis that contrasts this laser method with existing techniques.
- The study focuses on the removal process and short-term integrity of the underlying EMC but does not address whether the exposed EMC surface retains its mechanical and chemical stability over time. Include a brief discussion of the potential aging effects or propose future work to assess long-term reliability after EMI layer removal.
Author Response
Dear Reviewer
I am very appreciated for your valuable comments and constructive suggestions, which help us improve the quality of the manuscript. We attached the file of reviewer's comment and answers since this file contains the table and figures. Thank you very much.

Reviewer 2 Report
Comments and Suggestions for Authors
The paper entitled “Laser-Based selective removal of EMI shielding layers in advanced semiconductor packagess” studies a laser-based technique for selectively removing EMI shielding layers from advanced semiconductor packages without physical masking.
The paper is well-structured, clearly written and provides meaningful insights for applications in SiP and AiP technologies. However, there are several critical areas that require clarification or expansion before it is suitable for publication.
Here are some suggestions to improve to current text:
- The title is overly broad. “Advanced semiconductor packages” could refer to many systems beyond SiP; a more precise formulation is suggested.
- The abstract is good ; the objective is clearly stated, but more quantified results could be given.
- In the introduction, the study is a logical extension of known laser peeling techniques. A deeper comparison with alternative state-of-the-art methods (e.g., UV ablation, chemical stripping, dry etching) is lacking. As it is a hot topic, several more recent publications (from 2023) on EMI shielding removal or packaging delamination methods are missing and should be reviewed.
- In section ”Results and discussion”, the model assumes perfect bonding at all interfaces and does not account for real-world interface roughness or pre-existing micro-defects. This should be discussed in the argumentation. Also, the adhesion strength is not experimentally measured. Instead, delamination is inferred by comparing simulated stresses with literature values. The authors should provide quantitative or at least semi-quantitative validation of adhesion strength or justify why it cannot be measured.
- The technic described in the paper is convincing. However, it should be compared and contrasted with existing EMI removal methods in both performance and applicability. That would also provide a stronger conclusion.
Author Response
Dear reviewer.
We are greatly appreciated for your valuable comments and constructive suggestions, which help us to improve the quality of the manuscript. We attached the file of reviewer's comment and answers. Also based on your comments, we modified the manuscript.
Thank you very much
